# Estimating Long-Term Average Carbon Emissions from Fires in Non-Forest Ecosystems in the Temperate Belt

Andrey Ostroukhov [1], Elena Klimina [1], Viktoriya Kuptsova [1,*] and Daisuke Naito [2]

[1] Khabarovsk Federal Research Centre FEB RAS, Institute of Water and Ecology Problems FEB RAS, 56 Dikopoltseva, 680021 Khabarovsk, Russia; Ostran2004@bk.ru (A.O.); kliminaem@bk.ru (E.K.)
[2] Center for International Forestry Research, Bogor 16115, Indonesia; d.naito@cgiar.org
[*] Correspondence: Victoria@ivep.as.khb.ru; Tel.: +7-(4212)-325755

**Abstract:** Research into pyrogenic carbon emissions in the temperate belt of the Russian Federation has traditionally focused on the impact of forest fires. Nevertheless, ecosystems in which wildfires also make a significant contribution to anthropogenic $CO_2$ emissions are poorly studied. We evaluated the carbon emissions of fires in the non-forest ecosystems of the Middle Amur Lowland, in the Khabarovsk Territory of the Russian Federation. Our study is based on long-term Earth remote sensing data of medium spatial resolution (Landsat 5, 7, and 8) and expeditionary studies (2018–2021). The assessment of carbon directly emitted from wildfires in meadow and meadow–mire temperate ecosystems in the Middle Amur lowland shows that specific emissions from such ecosystems vary, from 1.09 t/ha in dwarf shrub–sphagnum and sphagnum–ledum and sedge–reed fens to 6.01 t/ha in reed–forb, forb, reed, and sedge meadows. Meanwhile, carbon emissions specifically from fires in meadow and meadow–mire ecosystems are less significant—often an order of magnitude less than carbon emissions from forest fires (which reach 37 tC/ha). However, due to their high frequency and the large areas of land burned annually, the total carbon emissions from such fires are comparable to annual emissions from fires in forested areas. The results obtained show that the inadequacy of the methods used in the automatic mapping of burns leads to a significant underestimation of the area of grassland fires and carbon emissions from non-forest fires.

**Keywords:** pyrogenic carbon emissions; remote sensing; non-forest ecosystems; Russia



## 1. Introduction

The dynamics of how the geosphere is transforming under the influence of human activity is a priority research area [1–3]. Climatic change studies are of particular importance [2–4], including those investigating wildfires' contribution to greenhouse gas emissions [5–11]. Since the 1970s, researchers have considered this area of research to be particularly significant [7,12], and since 1990, numerous studies have been published assessing the contribution of pyrogenic greenhouse gas emissions to total anthropogenic emissions [6,7,13–15]. Estimates now exist for carbon emissions from fires in tropical [15,16] and boreal forests [6,10,14,17–21], African [22–24] and Australian [25] savannas, and steppe regions [26,27]. Numerous works are likewise devoted to assessing the scale and impact of wildfires in the boreal zone of the Northern Hemisphere [10,19–21,28]; however, most authors focus primarily on forestry [10,14,19,20,29–31] and peat fires [32,33], as well as the consequences of agricultural burns in North America [34], Eastern Europe [34–37], and Asia [38].

In Russia, the largest forest state in the world, great attention has always been paid to studying the problems associated with forest fires. However, in the last decade, many works have appeared on fires in non-forest areas—the steppe and forest–steppe areas of the European part of Russia and the south of Western Siberia, and the steppes of the arid continental part of Eastern Siberia [39]. According to calculations by Vivchar, Moiseenko,

and Pankratova [10], non-forest fires in Russia for 2000–2008 accounted for 24–37% of the national contribution to $CO_2$ emissions. Studies carried out in the steppes of Russia (European part), Belarus, and Lithuania have revealed significant unexplained areas of burned agricultural land and pastures, including small areas, due to which the fire area has significantly increased [37]. At the same time, the European territory of Russia accounts for 31–36% of the world's agricultural burns [37].

Yet, such studies inadequately cover meadow and meadow–mire ecosystems, in which fires are characterized by low carbon emissions (when compared to forest ones) [28] but are very frequent due to the significant accumulation of combustible substances. Their characteristic feature is the presence of two 'fire hazard' seasons; this distinguishes meadows of the Russian Far East from Australian bush and African savannas, which are otherwise comparable in terms of the mass of combustible plant material [22,24]. The first period of herb drying is in spring, from the snowmelt until the beginning of active plant growth. The second is in autumn, from the end of the growing season until snow cover formation.

One such region, where temperate-zone meadow–mire ecosystems are widespread, is the Middle Amur Lowland (Figure 1). Within the Khabarovsk Territory, these ecosystems cover 4.116 mln ha. Large-scale wildfires occur here almost every year, spreading across meadow and meadow–mire ecosystems (Figure 2). These wildfires emerge due to the territory's climate, specific vegetation, and land use [40,41].

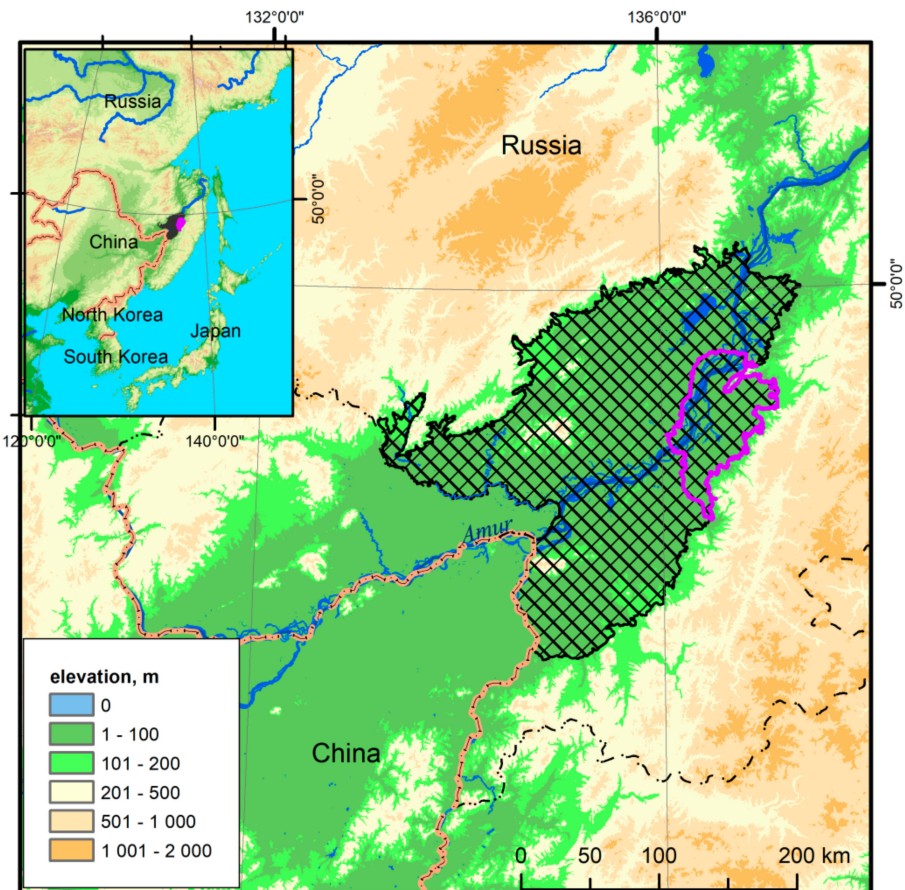

**Figure 1.** Geographical location of the Middle Amur Lowland. The area with black cross-hatching is the Middle Amur Lowland (MAL) within the Khabarovsk Territory, the purple outline identifies the key site of Lower Anyui (LA) studies, and the pink line represents state boundaries.

In spring, grass burns usually last no more than 3–4 weeks: fires start from the time of snow melting and the drying of the last year's dry grass (standing litter) in open areas and end with the beginning of active growth of herbaceous vegetation. Due to the flat position

and high degree of moisture, meadow and meadow–mire landscapes are widespread here. Significantly smaller areas are occupied by deciduous and small-leaved forests, growing in flat areas with good drainage and low elevation (Figure 3).

Active development of the Middle Amur Lowland began at the end of the 19th century but proceeded very unevenly; intensive economic activity at first covered only the western area, with the southern right-bank part included later. Currently, the total area of land transferred to economic activity (residential and industrial territories, reclaimed, and agricultural) is 7.6%, reflecting the low degree of development in the plain territory [40].

Previous studies looking at the dynamics of wildfires and pyrogenic greenhouse gas emissions in the Khabarovsk Territory [42] and in the south of Pacific Russia [19,43] were primarily aimed at assessing the consequences of forest fires. The impact of fires on non-forest lands in this region was previously studied in relation to the functioning of mire geosystems, depending on past and modern pyrogenesis [44,45]. No studies of the pyrogenic emissions from non-forest ecosystems in the territory of the Russian Far East have yet been carried out.

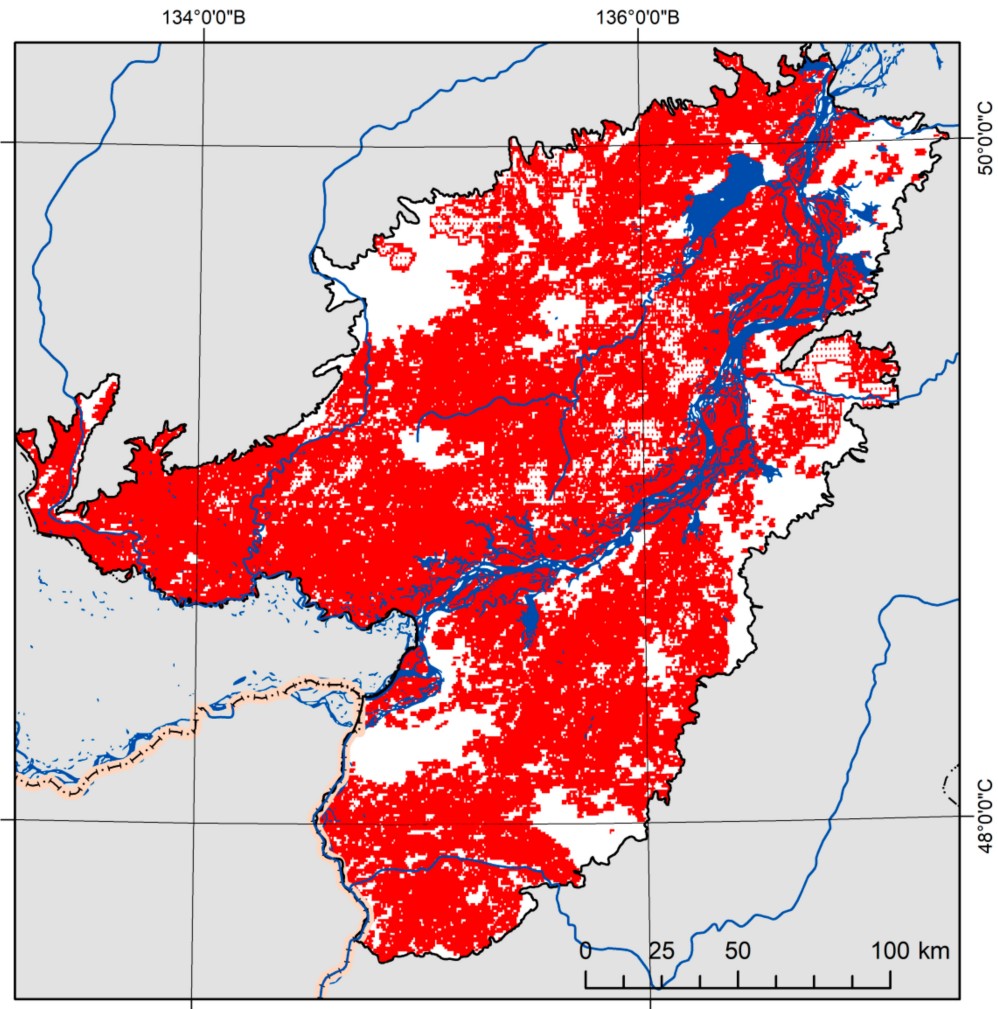

**Figure 2.** Fires in the Middle Amur Lowland (MAL) territory during 2000–2019 according to the Information System of Remote Monitoring of the Federal Forestry Agency of Russia [46].

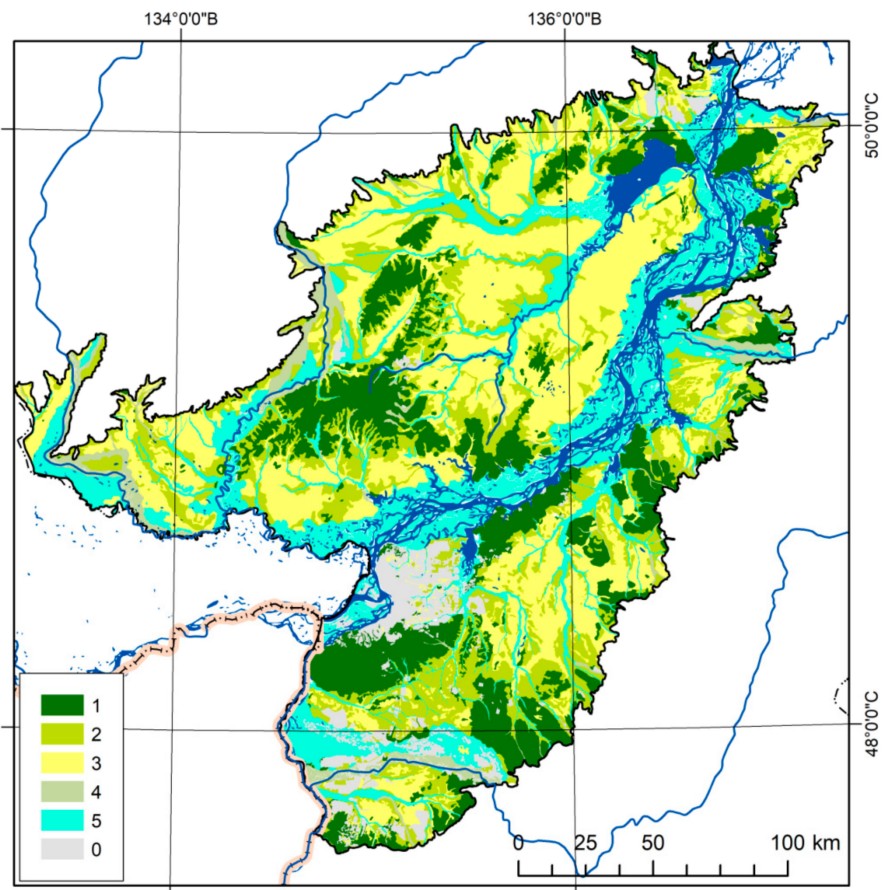

**Figure 3.** Types of ecosystems in the Middle Amur Lowland (MAL) territory. Types of ecosystems: 1, forest; 2, forest–meadow–mire; 3, meadow–mire; 4, floodplain, mainly forest; 5, floodplain, mainly meadow–shrub; 0, developed lands (residential and industrial, reclaimed, and agricultural) [40,47].

Therefore, the purpose of this study was to estimate carbon emissions from fires in non-forest ecosystems of the MAL (within the Khabarovsk Territory of the Russian Federation) based on Earth remote sensing data and materials from expeditionary studies.

## 2. Materials and Methods

### 2.1. Characteristics of the Study Area

The Middle Amur Lowland (MAL) is the northeastern part of the Sanjiang–Middle Amur Plain (a transboundary region that includes the northeastern part of the Chinese province of Suifenhe, the western part of the Jewish Autonomous Region, and the Khabarovsk Territory of Russia). Figure 1 shows its location within the Khabarovsk Territory of Russia (48–50° N, 134–137° E). The Amur River, one of the largest rivers in the world, crosses the plain from southwest to northeast. This territory is a low-lying accumulative plain, composed of alluvial deposits from the Neogene–Quaternary age, with individual remnants of low-altitude uplands belonging to the Cretaceous age [48,49]. To analyze the scale of pyrogenic impact on various ecosystems, the MAL ecosystem map developed by the authors [40] was used. Within the study area, four types of relief have been identified: floodplain, plain, foothill, and low mountain.

The MAL is characterized by the distribution of 49 subtypes of ecosystems grouped into 5 types. The forest types include spruce and fir, broad-leaved Korean pine, larch, and other forests (27 subtypes of ecosystems) occupying 18.5% of the lowland area (Figure 3). Forest–meadow–mire ecosystems are distributed across 19.4% of the MAL (8 subtypes of ecosystems). Floodplain ecosystems subdivided into forest (6 subtypes of ecosystems)

and meadow–shrub (3 subtypes of ecosystems) cover 27% of the territory. The flat relief combined with a sufficiently high moisture coefficient is responsible for the dominance of meadow–mire vegetation in the MAL at 30.1% (5 subtypes of ecosystems) (Figure 3).

Due to the vastness and inaccessibility of the study area within the MAL, the Lower Anyui (LA) key site was used for carrying out expeditionary (ground) research. This site reflected the entire diversity of the ecosystems, covering part of the Anyui River basin in the lower reaches (right tributary of the Amur River) and the adjacent territory over an area of 876 thousand hectares (Figure 1).

### 2.2. Estimating Carbon Emissions

Carbon emissions were estimated using the method proposed by Seiler and Crutzen (1980) (1):

$$E = A \times B \times FC \times ef,\tag{1}$$

where E is the carbon emission in tons per hectare (t/ha); A is the area burned by fire, in hectares; B is the biomass stock in tons per hectare (t/ha); FC is the carbon fraction of the biomass; and ef is the biomass combustion factor.

This method, which is used extensively globally [14,21,24,36], has many variations to account for the specifics of various ecosystems [14,28] and regional distribution patterns of fires [20,24,31].

This formula for calculating emissions is well suited for use in the MAL territory as it considers two important aspects: first, the specific structural and spatial features of ecosystems (e.g., their high mosaicity), and second, the fact that there is a large proportion of mires in this area (the formula allows one to take into account the carbon emission when the moss cushion burns out). The high mosaicity of the spatial structure of the territory's ecosystems makes it impossible to clearly separate them into types. For example, within the MAL, 6% of the territory is occupied by mixed herbs, reed grass, sedge meadows in combination with fens (along the depressions), and oak–white birch–aspen belt forests (along the elevated areas) that are not divided into separate categories according to remote sensing data. This requires taking into consideration the areas of individual vegetation in the previously identified types of ecosystems based on aerial photography data from the DJI Phantom 4 UAV (for each fieldwork site described) (Figure 4).

To reveal the spatial distribution of carbon stocks, calculations were carried out for the identified subtypes of ecosystems within the LA key site. For each of them, the proportions of forest, meadow, and mire communities were determined, which were then taken into account when calculating carbon emissions (2, 3).

As a result, the formula took the following form:

$$E = A \times SE,\tag{2}$$

where E is the carbon emission in tons, SE is the specific carbon emission in tons per hectare (t/ha), and A is the area of a natural fire, in hectares.

$$SE = F \times SE_f + M \times B_m \times FC_m \times ef_m + S \times B_s \times FC_s \times ef_s + S \times N \times B_n \times FC_n \times ef_n \tag{3}$$

Here, B is the biomass reserves in tons per hectare (t/ha), FC is the carbon fraction of the biomass, ef is the biomass combustion factor, $SE_f$ is the specific carbon emission of forest in tons per hectare (t/ha), F is the share of forest communities, M is the share of meadow communities, S is the share of mire communities (based on the Phantom 4 UAV data from for each site to be described during field survey), and N is the share of burnt sphagnum sod. The FC value for the biomass was taken to be equal to 0.5 [20,31], while B and ef were taken from literature data [10,20,21,31,50–52]. It should be noted that biomass-to-carbon conversion coefficient (FC) values from 0.42 to 0.5 have been reported in the literature sources concerning carbon emissions or sinks [53–55]. Indeed, an overall average conversion coefficient of 0.5 is widely used for forest ecosystems. However, the carbon

content in sedges ranges from 47.9 to 50.8%, that in reed ranges from 48 to 49.8%, that in cotton grass ranges from 49.1 to 50.0%, that in Scheuchzeria is around 51.1%, and that in ericaceous dwarf shrubs (cranberry, ledum, Andromeda) ranges from 53.5 to 53.9% [56,57]. After analyzing these data, we decided on a conversion coefficient of 0.5. Calculations of carbon emissions were carried out for each natural unit for two seasons of high burn frequency (spring and autumn).

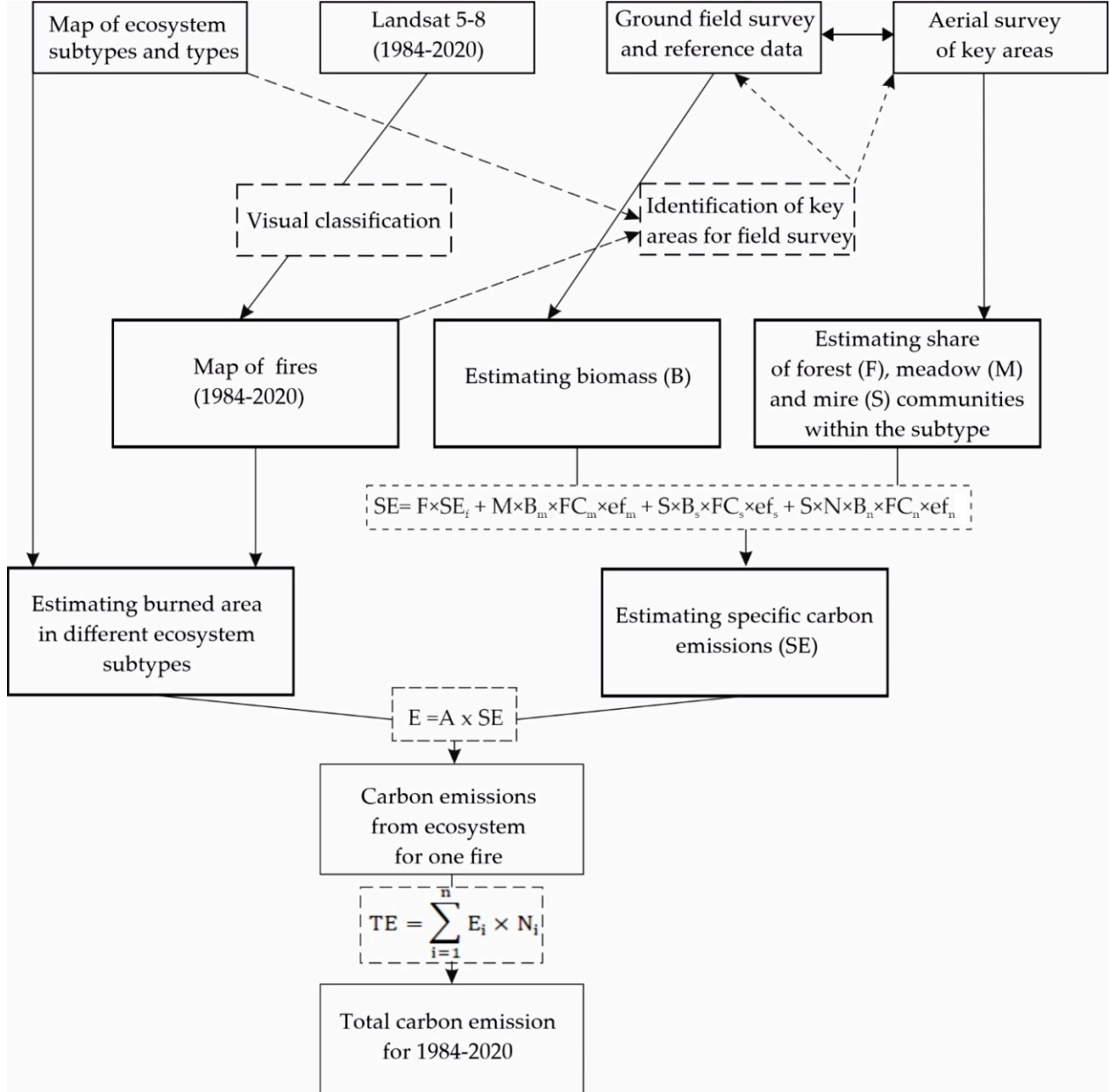

**Figure 4.** Flow diagram of the carbon emissions calculations for the ecosystem.

The total emissions (TE) for 1984–2020 were determined as the sum of emissions E from each type of ecosystem, calculated using the formula:

$$TE = \sum_{i=1}^{n} E_i \times N_i \tag{4}$$

where $E_i$ is the emissions of a particular type of ecosystem, and $N_i$ is the number of fires over the period.

For forest ecosystems, published data on specific carbon emissions were used [20,31,50]. Data on the aboveground biomass stocks of meadow and mire ecosystems were collected during expeditionary studies in April–May 2018 and 2021. For this, reference plots were selected, on which a complete geobotanical description of the vegetation cover was carried out. To determine the mass of dry combustible material in the studied ecosystem types, the dry standing litter of herbaceous plants was cut from an area of 1 m2 in three replicates. Dry branches of shrubs and monoliths (10 cm × 10 cm × 30 cm) of dominant species of sphagnum mosses were also selected from the mires for subsequent determination of the mass of burnt mosses. In burned areas, the proportion of burnt sphagnum sods (as a percentage) and the depth of burning were determined. The selected plant biomass was sorted by species, dried to a completely dry mass, and weighed. In the MAL area, fires take place mainly in the spring, when seasonal permafrost remains, which prevents the fire from spreading deep into the peat and soil. The high speed of fire propagation does not allow the fire to spread deeply. Therefore, carbon emissions from peat and soil combustion were not estimated. The data obtained at the LA key site were used to calculate pyrogenic emissions throughout the MAL area.

### 2.3. Estimating the Burned Area

Two main types of remote sensing data are currently used to assess the area impacted by fire:

1. Active combustion is detected using low-spatial-resolution data (250/500/1000 m) with a high sampling rate. These data include information obtained by MODIS (Terra and Aqua satellites) and VIIRS (NPP satellite) [23,24,35,36,58].
2. Analyzing the consequences of burning, and the mapping of burns, uses data of medium and high spatial resolution (10/15/30 m) in the visible and infrared range, which allows for detailed spatial assessments [21,36,59–61].

In this work, the second approach was used to highlight the areas covered by fire. This choice is due to the fact that the remote sensing data of active combustion, obtained by MODIS devices, are characterized by a low spatial resolution. This has resulted in previous researchers noting challenges detecting fires [17,53,61], due to narrow fire edges (where the fire edge appears in the form of a narrow and long strip), small combustion zone areas, and strong smoke all preventing fire detection within the meadow and meadow–mire ecosystems.

The high frequency at which grass fires occur (annually, and sometimes twice a year), and the alternation of years with low and high fire rates, meant that a long observation period was needed to determine the average long-term characteristics of wildfires and their trends [24,60]. The high rate of vegetation renewal in the spring and a relatively low frequency of satellite observations hamper the identification of burns in herbaceous ecosystems. Therefore, all cloudless data of free access from Landsat satellites 5 (475 pcs.), 7 (328 pcs.), and 8 (189 pcs.) for the period from 1984 to 2020 were obtained (Table 1) from the Earth Explorer website (https://earthexplorer.usgs.gov/, accessed on 30 January 2022). Remote sensing data processing was carried out using a visual method in the QGIS 3.18.1 program to contour the territories affected by fires in the spring and autumn seasons (separately) for each year. When spring fires were identified, areas impacted by the previous autumn's fires were removed, to avoid any decryption errors.

**Table 1.** Numbers of images used from various satellites.

| Year | Landsat-5 | Landsat-7 | Landsat-8 | Year | Landsat-5 | Landsat-7 | Landsat-8 | Year | Landsat-5 | Landsat-7 | Landsat-8 |
|---|---|---|---|---|---|---|---|---|---|---|---|
| 1984 | 7 | – | – | 1997 | 23 | – | – | 2010 | 11 | 17 | – |
| 1985 | 11 | – | – | 1998 | 18 | – | – | 2011 | 8 | 19 | – |
| 1986 | 10 | – | – | 1999 | 15 | – | – | 2012 | – | 23 | – |
| 1987 | 20 | – | – | 2000 | 14 | 18 | – | 2013 | – | 23 | 24 |
| 1988 | 16 | – | – | 2001 | 22 | 17 | – | 2014 | – | 28 | 27 |
| 1989 | 19 | – | – | 2002 | 11 | 12 | – | 2015 | – | 16 | 16 |
| 1990 | 19 | – | – | 2003 | 15 | 8 | – | 2016 | – | 21 | 19 |
| 1991 | 14 | – | – | 2004 | 18 | 14 | – | 2017 | – | 1 | 25 |
| 1992 | 23 | – | – | 2005 | 12 | 18 | – | 2018 | – | – | 27 |
| 1993 | 17 | – | – | 2006 | 19 | 19 | – | 2019 | – | 13 | 28 |
| 1994 | 22 | – | – | 2007 | 22 | 18 | – | 2020 | – | – | 23 |
| 1995 | 17 | – | – | 2008 | 23 | 19 | – | | | | |
| 1996 | 23 | – | – | 2009 | 26 | 24 | – | Total | 475 | 328 | 189 |

## 3. Results

### 3.1. Estimating the Wildfire-Impacted Land Area in the Middle Amur Lowland

Processing long-term series of Earth remote sensing data for the Middle Amur Lowland (MAL) site allowed us to identify areas burned by fires in spring and autumn during 1984–2020 (Table 2, Figure 5). Over this period, the smallest pyrogenic impacts were observed in 1984, 1994, and 2010, when fires burned 5.7%, 5.3%, and 2.8% of the MAL territory, respectively; in the most unfavorable years (1996 and 2005), fire-burned areas exceeded 50% of the total land area. Considering the whole time period, an average of 25.4% of the total land area was impacted by fires.

A spatial analysis of the data obtained enabled us to assess the frequency of wildfires in different types of vegetation. It was revealed that significant areas of MAL were exposed to fire many times—from 2 to 36 times over the course of 37 years (Table 3, Figure 6). The total area impacted by fires during this period amounted to more than 38 mln ha—938% of the total land area in the Middle Amur Lowland (Table 3). Forest fires accounted for only 12% of this. Meanwhile, the scale and frequency of fires that are typical in meadow and meadow–mire ecosystems in floodplains meant that 1,317.4% of this ecosystem's total land area was impacted by fire.

**Table 2.** Fire areas within the Middle Amur Lowland during 1984–2020.

| Year | Area, Thousand ha | Share of the Total Area, % | | | Year | Area, Thousand ha | Share of the Total Area, % | | |
|---|---|---|---|---|---|---|---|---|---|
| | | Spring | Autumn | Total | | | Spring | Autumn | Total |
| 1984 | 221.23 | 5.7 | 0.0 | 5.7 | 2003 | 1445.17 | 36.6 | 0.5 | 37.1 |
| 1985 | 354.14 | 7.7 | 1.4 | 9.1 | 2004 | 650.75 | 14.9 | 1.8 | 16.7 |
| 1986 | 789.80 | 17.5 | 2.7 | 20.3 | 2005 | 2262.88 | 8.5 | 49.6 | 58.1 |
| 1987 | 1849.10 | 47.4 | 0.1 | 47.5 | 2006 | 807.33 | 17.1 | 3.6 | 20.7 |
| 1988 | 402.72 | 6.8 | 3.5 | 10.3 | 2007 | 697.98 | 11.5 | 6.5 | 17.9 |
| 1989 | 1224.59 | 31.2 | 0.3 | 31.4 | 2008 | 1547.11 | 35.5 | 4.2 | 39.7 |
| 1990 | 527.20 | 13.0 | 0.6 | 13.5 | 2009 | 1754.71 | 42.9 | 2.2 | 45.1 |
| 1991 | 389.19 | 10.0 | 0.0 | 10.0 | 2010 | 110.96 | 1.7 | 1.1 | 2.8 |
| 1992 | 521.39 | 8.9 | 4.5 | 13.4 | 2011 | 508.61 | 12.2 | 0.9 | 13.1 |
| 1993 | 1252.84 | 32.0 | 0.2 | 32.2 | 2012 | 752.91 | 19.0 | 0.4 | 19.3 |
| 1994 | 205.16 | 4.5 | 0.8 | 5.3 | 2013 | 670.58 | 16.7 | 0.6 | 17.2 |
| 1995 | 1070.77 | 25.1 | 2.4 | 27.5 | 2014 | 1625.70 | 34.9 | 6.9 | 41.7 |
| 1996 | 2335.10 | 58.6 | 1.4 | 60.0 | 2015 | 704.13 | 14.4 | 3.7 | 18.1 |
| 1997 | 854.52 | 20.2 | 1.7 | 21.9 | 2016 | 1790.69 | 14.4 | 31.6 | 46.0 |
| 1998 | 1190.61 | 17.1 | 13.5 | 30.6 | 2017 | 426.15 | 8.8 | 2.1 | 10.9 |
| 1999 | 1202.35 | 29.5 | 1.4 | 30.9 | 2018 | 926.45 | 23.5 | 0.3 | 23.8 |
| 2000 | 1016.17 | 21.9 | 4.2 | 26.1 | 2019 | 1444.55 | 36.8 | 0.3 | 37.1 |
| 2001 | 1719.17 | 19.4 | 24.7 | 44.1 | 2020 | 419.32 | 9.4 | 1.4 | 10.8 |
| 2002 | 872.33 | 22.0 | 0.4 | 22.4 | Mean | 987.68 | 20.5 | 4.9 | 25.4 |

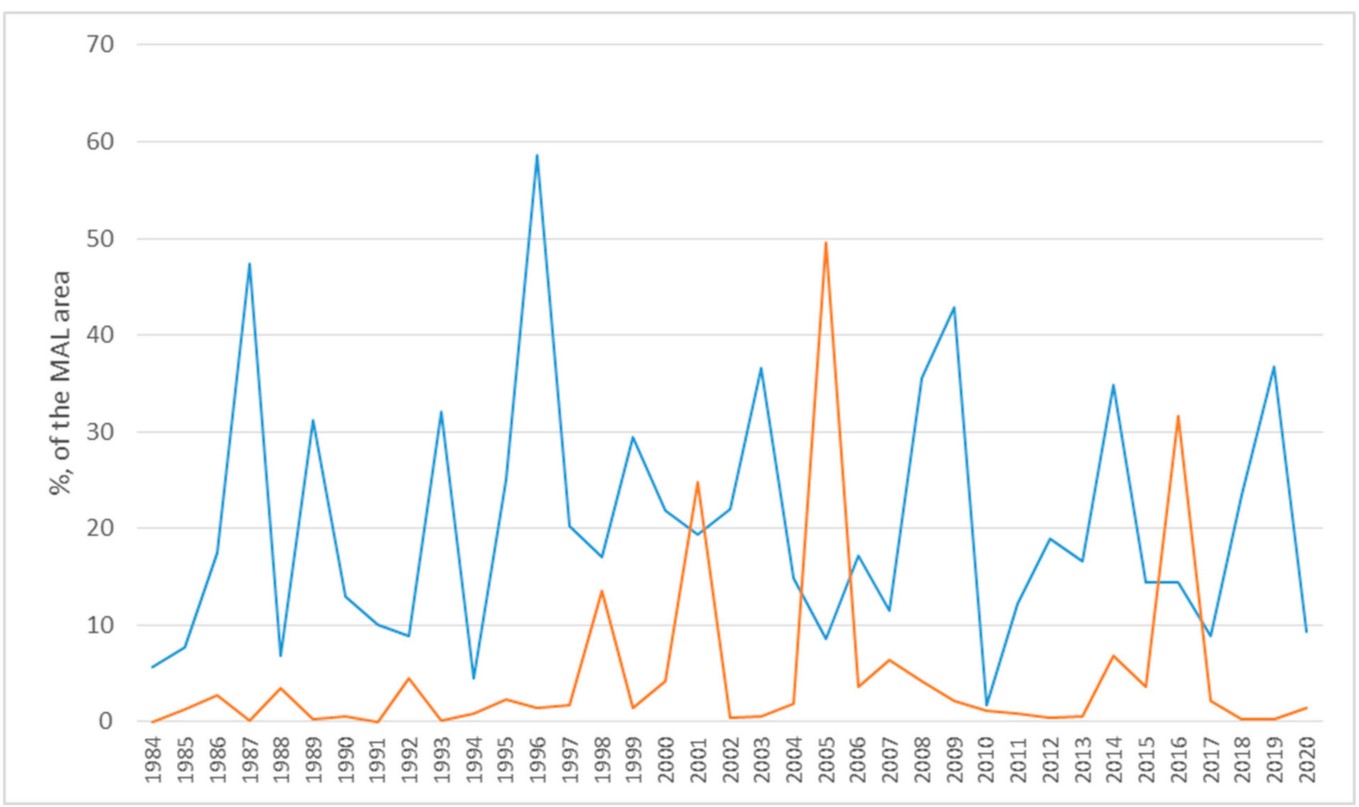

**Figure 5.** Areas of spring (blue line) and autumn (red line) fires in the Middle Amur Lowland territory during 1984–2020.

**Table 3.** Burned areas in different ecosystems in the Middle Amur Lowland (1984–2020).

| Eco-System Type [1] | Ecosystem Area | | Share of Ecosystem not Affected by Fires, % | Burnt Forest Area | | Share of Spring Fires, % | Average Annual Burnt Forest Area | |
|---|---|---|---|---|---|---|---|---|
| | Total, Thousand ha | % of the MAL Area | | Total, Thousand ha | % of the Ecosystem Area | | Total, Thousand ha | % of the Ecosystem Area |
| 1 | 717.49 | 18.5 | 22.82 | 3194.09 | 445.17 | 76.86 | 86.33 | 12.03 |
| 2 | 754.06 | 19.4 | 5.46 | 7967.6 | 1056.62 | 78.50 | 215.34 | 28.56 |
| 3 | 1165.98 | 30.1 | 4.09 | 11,659.86 | 1000.00 | 77.30 | 315.13 | 27.03 |
| 4 | 165.00 | 4.3 | 36.75 | 732.08 | 443.68 | 85.33 | 19.79 | 11.99 |
| 5 | 881.86 | 22.7 | 2.72 | 11,617.5 | 1317.39 | 85.67 | 313.97 | 35.61 |
| ∑ | 3894.25 | 100.0 | 10.20 | 36,542.73 | 938.38 | 80.67 | 987.64 | 25.36 |

[1.] Ecosystem type: 1, forest; 2, forest–meadow–mire; 3, meadow–mire; 4, floodplain, mainly forest; 5, floodplain, mainly meadow–shrub.

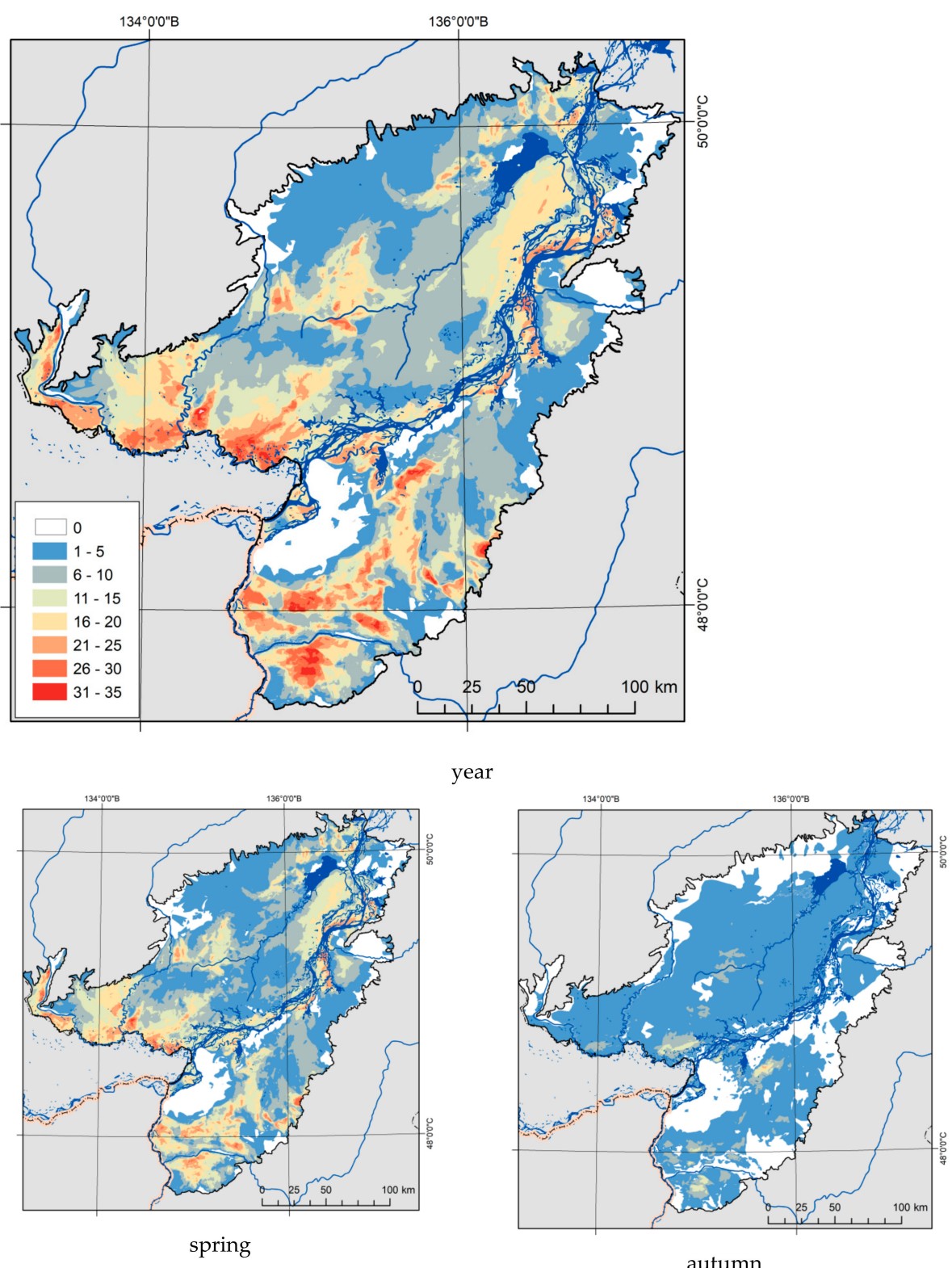

year

spring

autumn

**Figure 6.** Number of fires in the Middle Amur Lowland between 1984 and 2020. The black line indicates the border of the MAL.

As already noted, the highest number of fires in natural ecosystems in the south of the Russian Far East is observed in spring [19]. According to long-term statistics, about half of all forest fires (49.2%) occur at this time. In the summer period, 36.9% of the total recorded

number of fires occur, usually associated with prolonged droughts, with autumn seeing no more than 13.5% of the fires [62].

In meadow–mire and floodplain meadow ecosystems, fires are even more confined to the spring and autumn seasons. Table 3 shows that spring burns within the MAL account for 80.7% of the fire-impacted area and mainly affect non-forest lands; autumn burns account for 19.3%; while no fires were recorded during the summer. Individual peaks of autumn fires in 2001, 2005, and 2016 were associated with a significant accumulation of combustible materials following the dry and warm autumn of the previous year.

On average, the cycle of meadow fires is 2–4 years, which is associated with the accumulation of combustible material in natural ecosystems (mainly grasses) [26]. Over most of the MAL, fires repeated 4–10 times between 1984 and 2020; however, in some areas, the frequency of fires was as much as 36 times in these 37 years (Figure 6). Areas where fires were most frequently seen were also the most developed: urban and rural settlements with well-developed transport infrastructure and agricultural lands (south-western and southern parts of the MAL) and areas adjacent to the railway and major highways.

*3.2. Estimating Carbon Emissions*

During 2018–2019, field studies were carried out within the key LA site. In the course of the work, data were collected on reserves of dry combustible matter in the meadow and mire ecosystems typical of the MAL territory. In total, 60 plots were laid within 20 key areas located in various subtypes of ecosystems that were affected by fires over the last 3 years (the fire age was specified according to remote sensing data). The combustible materials (grasses, sedges, shrubs, and mosses) were sampled, and a geobotanical description and aerial photography were carried out at each plot using the methods described above (Section 2.2. Estimating Carbon Emissions). After analyzing field data and published materials, based on the adapted model of Seiler and Crutzen [9]), we were able to calculate both the carbon emissions specifically emitted during fires across various types of ecosystems (2, 3) and the total emissions (4) for 1984–2020 (Table 4, Figure 7).

According to the results, the average annual emissions from the MAL territory amount to 2.68 million tons of carbon (91.16 million tons of carbon over 37 years). At the same time, forest ecosystems, despite their high SE values (up to 37 t/ha), do not significantly affect the total carbon emissions generated from the MAL territory. This is explained, among other things, by the fact that in the spring and autumn periods, forest fires do not have a high intensity and do not affect the tree layer [20]. At the same time, over 52% of total emissions (TE) is associated with wildfires in floodplain meadow ecosystems, and about 29.4% is associated with wildfires in meadow–mire ecosystems.

**Table 4.** Carbon emissions from the Middle Amur Lowland (MAL) territory for 1984–2020.

| Eco-System Type [1] | Area, ha | Area,% of the MAL | SE [2] for 1 Fire, t/ha | Range of SE Values for one Fire, t/ha | E [3] for 1984–2020, t | E [3] for 1984–2020, % of total | E [3] Spring for 1984–2020, % of the Total | Average Long-Term SE [2] from 1 ha, t/ha | Range of Mean Long-Term SE [2] Values from Each Ecosystem per Year, t/ha |
|---|---|---|---|---|---|---|---|---|---|
| 1 | 717,497.26 | 18.5 | 1.21 | 0.5–37 | 4,069,594.67 | 4.5 | 76.5 | 0.15 | 0–3.34 |
| 2 | 754,063.06 | 19.4 | 1.68 | 1.1–2.8 | 12,225,227.30 | 13.4 | 79.0 | 0.44 | 0.17–0.93 |
| 3 | 1,165,982.94 | 30.1 | 2.20 | 1.1–2.8 | 26,796,268.70 | 29.4 | 77.6 | 0.62 | 0.02–0.89 |
| 4 | 165,003.37 | 4.3 | 0.61 | 0.3–1.5 | 566,417.39 | 0.6 | 84.5 | 0.09 | 0.02–0.68 |
| 5 | 881,856.97 | 22.7 | 3.66 | 1.4–6.0 | 47,508,857.41 | 52.1 | 86.5 | 1.46 | 0.39–2.51 |
| ∑ | 3,894,246.50 | 100.0 | 2.25 | 0.4–6.0 | 91,166,365.47 | 100.0 | 82.4 | 0.63 | 0–2.51 |

[1]. Ecosystem type: 1, forest; 2, forest–meadow–mire; 3, meadow–mire; 4, floodplain, mainly forest; 5, floodplain, mainly meadow–shrub. [2]. SE, specific carbon emission; [3]. E, carbon emission.

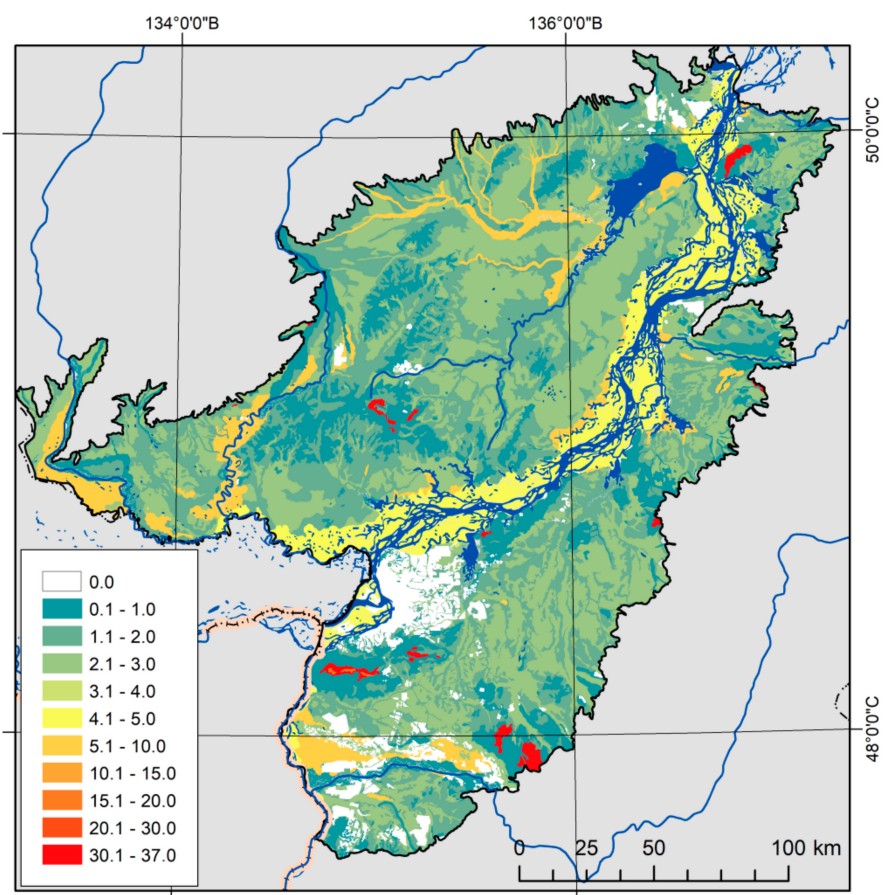

**Figure 7.** Specific carbon emissions from various ecosystems in the MAL territory (t/ha).

Due to significant differences in the size of areas in which fires occurred in different years, the value of the average long-term specific emission (SE), calculated as the ratio of the long-term emission of ecosystems of a certain type to the total area of this ecosystem type, is more informative. It shows the average long-term SE values from 1 ha of ecosystems of various types, taking into account the average long-term fire rate. For forest ecosystems, the average long-term SE is 8 times lower than the SE, while for floodplain meadows, the average long-term SE is 2.5 times lower than the SE, reflecting their higher frequencies of burning (Table 4).

## 4. Discussion

### 4.1. Estimating the Burned Area

As already noted in the Introduction, when assessing the scale of fires across the Russian Federation and the Far East of Russia, most authors mean primarily forest fires. This is confirmed by official statistical data, according to which non-forest fires account for 22% in Russia and 8.9% in the Khabarovsk Territory [46].

Nevertheless, according to the results of a project that mapped landscape fires across the Russian Federation in 2020 [63], the area of fires taking place in spring accounts for more than half—52%—of all those that happen per year (13.5 mln ha) for the territory of Russia and 76.7% of those for the Khabarovsk Territory. As well as forest fires, this prior work took into account grass, reed, peat, tundra fires, and agricultural burns. According to these data, the share of spring fires in the MAL territory was 82.4%, which is comparable to the results of the current study (Figure 6).

To check the reliability of the cartographic materials, we compared our results with the global annual burned (forest) area maps (GABAM), with a resolution of 30 m [59], and the data on the number of hotspots (HS) from the Fire Information for Resource Management

System website [64] and Visible Infrared Imaging Radiometer Suite (VIIRS) for the years 2012, 2016, 2018, and 2020.

Figure 8 reflects a significant underestimation of the areas of fires, both based on the materials of mapping burns in an automatic mode using Landsat-8 data (GABAM) and based on hotspot (HS) detection (VIIRS), which was also noted by various other authors [10,23,29,36,60–67]. However, the results of this study are close to the data obtained by the expert interpretation of materials of medium and high spatial resolution (Sentinel-2). The burnt area in the spring of 2020, according to the obtained data, amounted to 349.5 ths ha, and according to Greenpeace data, it was 354.5 ths ha. The slight difference is explained by the fact that agricultural fires were not taken into account in this work.

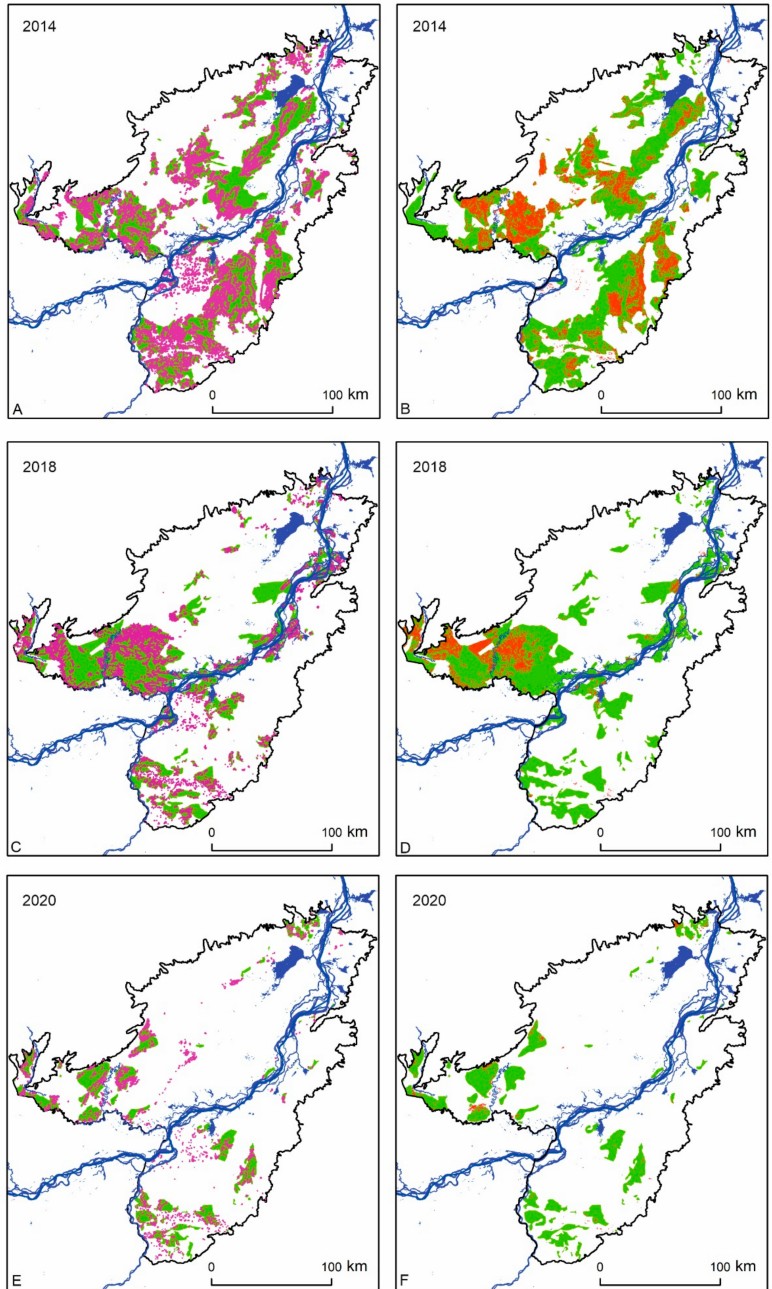

**Figure 8.** Burned areas within the Middle Amur Lowland (MAL) territory during 2014, 2018, and 2020 according to various data sources: VIIRS (**A**,**C**,**E**), GABAM (**B**,**D**,**F**) [59], and the authors' data (green).

Possible reasons for underestimating fire areas are as follows:

1. Smoke screens preventing the observation of burns [58,59,61];
2. Small stocks of combustible plant materials, leading to low combustion rates [35,36,58];
3. The impossibility of detecting fires with low-spatial-resolution satellite devices, due to their small area or the high speed of the fire edge, which appears in the form of a narrow and long strip [29,34,37,68,69];
4. Rapid regeneration and growth of herbaceous vegetation in the spring, reducing the radiation in the infrared channel and making it difficult to automatically classify burns [59];
5. Heterogeneity of the land cover leading to a large range of background temperatures and complicating the selection of hotspots [29].

However, it should be noted that the expert interpretation technique used in this study requires a significant investment of time and is inapplicable for rapid assessments of the impact of fires [65].

An analysis of the spatial distribution of fires within the MAL territory showed a convergence between areas that have frequently recurring fires and developed territories and transport infrastructure facilities. In addition to agricultural land use, fires are caused by hunting, fishing, and harvesting wild plants. This is related to the high frequency of fires along rivers and lakes in the northern and central parts of the plain. These rivers and lakes play the role of transportation arteries. All this points to human-started wildfires within the MAL, as confirmed by official statistical data [70] and the work of other researchers [19].

### 4.2. Estimating Carbon Emissions

We estimated the average long-term emissions from the MAL territory to be 2.68 million tC or 0.63 tC/ha. This appears to be relatively small compared to emissions from more productive forest biomes (Table 5), but it is comparable to the contribution of grass and shrub ecosystems in other regions of the world.

**Table 5.** Carbon emissions from fires in different regions, by ecosystem, according to Hoelzemann et al. [71].

| Region | Burnt Forest Area, $10^6$ ha | Total Carbon Emission, TgC | Carbon Emission, tC/ha | | |
| --- | --- | --- | --- | --- | --- |
| | | | Grasslands | Woodlands | Forests |
| North America | 7.0 | 196.1 | 4.3 | 16.7 | 29.2 |
| Central America | 2.0 | 43.7 | 1.8 | 6.6 | 27.6 |
| South America | 12.7 | 126.5 | 2.4 | 9.2 | 39.1 |
| North Africa | 60.4 | 408.7 | 1.4 | 7.6 | 34.6 |
| South Africa | 57.7 | 472.6 | 1.5 | 7.4 | 41.1 |
| Western Europe | 0.3 | 3.5 | 3.8 | 13.8 | 17.8 |
| Eastern Europe | 1.0 | 11.9 | 6.2 | 25.1 | 24.5 |
| North and Central Asia | 8.8 | 321.6 | 9.1 | 35.0 | 41.3 |
| Middle East Asia | 0.8 | 5.4 | 2.6 | 8.7 | 23.3 |
| East Asia | 0.0 | 0.1 | 2.3 | 12.2 | 22.1 |
| South Asia | 3.6 | 99.7 | 4.7 | 15.7 | 57.3 |
| Oceania | 17.8 | 51.6 | 1.0 | 3.8 | 30.0 |
| Middle Amur Lowland (MAL) | 1.0 | 2.68 | 1.1–6.0 | 1.1–2.8 | 0.5–37.0 |

Based on data on the distribution of burns in Russia in 2020 [63], maps of terrestrial ecosystems in the Russian Federation [47], and the results of the current work, it is possible to roughly estimate the carbon emissions of wildfires for some types of ecosystems in Russia. As can be seen in Table 6, carbon emissions from forest fires in 2020 amounted to 58.46 million tC, while emissions from meadow ecosystems alone were 9.9 million tC, or 16.9% of the E emissions from forest areas.

**Table 6.** Distribution of burned areas [1] and specific carbon emissions [2,3] from different types of ecosystems in Russia [4] in 2020.

| Ecosystems | Ecosystem Area | | Fire Area | | | Specific Emission (SE), t/ha | Emission, t |
|---|---|---|---|---|---|---|---|
| | ha | % of the Total Area | ha | % of the Total Area | % of the Ecosystem Area | | |
| Forest | 774,096,314 | 45.66 | 10,328,466 | 40.2 | 1.33 | 5.66[3] | 58,459,119 [3] |
| Shrubs | 32,253,309 | 1.90 | 329,085 | 1.3 | 1.02 | | |
| Needle-leaf evergreen shrubs | 25,155,249 | 1.48 | 193,547 | 0.8 | 0.77 | | |
| Broadleaf deciduous shrubs | 7,098,060 | 0.42 | 135,538 | 0.5 | 1.91 | 1.09 [2] | 147,736 [2] |
| Wetlands | 70,184,473 | 4.14 | 588,852 | 2.3 | 0.84 | | 1,422,333 [2] |
| Bogs and marches | 56,032,970 | 3.31 | 425,793 | 1.7 | 0.76 | 2.20 [2] | 936,744 [2] |
| Palsa bogs | 14,151,503 | 0.83 | 139,839 | 0.5 | 0.99 | 2.79 [2] | 390,151 [2] |
| Riparian | 7,888,022 | 0.47 | 23,221 | 0.1 | 0.29 | 4.11 [2] | 95,438 [2] |
| Herbaceous | 69,496,750 | 4.10 | 4,848,795 | 18.9 | 6.98 | | |
| Humid grasslands | 44,641,280 | 2.63 | 4,059,132 | 15.8 | 9.09 | 2.44 [2] | 9,904,282 [2] |
| Steppe | 24,855,469 | 1.47 | 789,664 | 3.1 | 3.18 | | |
| **Tundra** | **321,834,343** | **18.98** | **1,070,629** | **4.2** | **0.33** | | |
| Sedge tundra | 73,639,133 | 4.34 | 388,771 | 1.5 | 0.53 | | |
| Shrub tundra | 158,263,028 | 9.34 | 609,292 | 2.4 | 0.38 | | |
| Prostrate shrub tundra | 89,932,183 | 5.30 | 72,567 | 0.3 | 0.08 | | |
| **Complexes** | **208,418,307** | **12.29** | **7,283,982** | **28.3** | **3.49** | | |
| Recent burns | 12,506,668 | 0.74 | 229,698 | 0.9 | 1.84 | | |
| Croplands | 107,894,078 | 6.36 | 3,829,982 | 14.9 | 3.55 | | |
| Forest–Natural Vegetation complexes | 22,644,624 | 1.34 | 305,637 | 1.2 | 1.35 | | |
| Forest–Cropland complexes | 22,867,109 | 1.35 | 744,608 | 2.9 | 3.26 | 1.68 [2] | 1,250,941 [2] |
| Cropland–Grassland complexes | 42,505,828 | 2.51 | 2,174,056 | 8.5 | 5.11 | | |
| **Total** | **1,695,368,583** | **100.0** | **25,716,122** | **100** | **1.52** | | |

[1.] Database of the distribution of burns in Russia in 2020 [63]. [2.] Data from this research. [3.] Ershov and Sochilova [72]. [4.] Map of terrestrial ecosystems of the Russian Federation [47].

Within the Khabarovsk Territory, the specific emission (SE) value can reach 37 tC/ha during a forest fire. At the same time, it takes 20–50 years for the primary restoration of forest vegetation in areas affected by fire, so they can return to the state of a closed-canopy forest [73]. In meadow–mire ecosystems, the SE indicator is much lower (up to 6 t/ha), but the rapid accumulation of combustible materials (primarily herbaceous vegetation) means that wildfires are frequent and often annual.

## 5. Conclusions

Despite the large number of works devoted to the pyrogenic emission of greenhouse gases into the atmosphere, at present, this issue, in our view, remains insufficiently studied for some types of ecosystems. One of these biomes is the humid meadows of the temperate belt.

The results of this study show that official statistics significantly underestimate the areas of meadow fires [46]. According to long-term data, fires affect, on average, between 27% and 35% of non-forest ecosystems within the Middle Amur Lowland; in some years, over 50% of the non-forest area is impacted by these fires, comparable to the land area impacted by forest fires in the Khabarovsk Territory as a whole. Yet, according to official data, just 8.9% of the fires in the Khabarovsk Territory are non-forest fires.

Based on expedition data, it was possible to estimate the direct carbon emissions of wildfires in meadow and meadow–mire temperate ecosystems in the Middle Amur Lowland (MAL). The specific carbon emissions from such ecosystems vary from 1.09 t/ha in dwarf–sphagnum, sphagnum–ledum, and sedge–reed fens to 6.01 t/ha in reed–forb, forb, reed, and sedge meadows. Over the course of 1984 to 2020, the average annual emissions

from fires in natural ecosystems in the MAL territory were estimated at 2.68 million tC or 0.63 tC/ha.

Fire-specific carbon emissions from meadow and meadow–mire ecosystems are small (6.0 tC/ha) and are often an order of magnitude less than emissions from forest fires (which reach 37 tC/ha). However, due to the large extent of areas impacted and high frequency of burns annually, the total carbon emissions from meadow and meadow–mire fires are comparable to the annual emissions from fires in forested areas.

The influence of meadow fires is underestimated, primarily due to inadequacies in the methods used to automatically map burned areas, leading to an underestimation of their areas.

**Author Contributions:** Conceptualization, A.O.; methodology, A.O.; software, A.O.; validation, A.O.; formal analysis, A.O.; investigation, A.O., E.K. and V.K.; writing—original draft preparation, A.O.; writing—review and editing, A.O., E.K., V.K. and D.N. All authors have read and agreed to the published version of the manuscript.

**Funding:** This research was partially supported by the Center for International Forestry Research (CIFOR), the project title "Tropical Forest Fires to Promote Sustainable Rural Livelihoods".

**Data Availability Statement:** The data presented in this study are available on request from the corresponding author. The data are not publicly available due to privacy restrictions.

**Conflicts of Interest:** The authors declare no conflict of interest. The funders had no role in the design of the study; in the collection, analyses, or interpretation of data; in the writing of the manuscript; or in the decision to publish the results.

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
