# Peer review of "Estimating Long-Term Average Carbon Emissions from Fires in Non-Forest Ecosystems in the Temperate Belt"

_remotesensing, doi:10.3390/rs14051197_

Round 1

Reviewer 1 Report

The revisions made by the authors cover the issues raised in the first review. Therefore is recommended for publication

Author Response

Dear reviewer!

We would like to thank you for your valuable comments, and we changed the manuscript according them.

The introduction contains a wealth of references, but putting 53 references in a single paragraph does allow an accurate assessment of their relevance and critical findings.

Corrected

The most important paragraph of Section 1, 2nd in p 2, is necessary to be revised so that innovative elements of the work are highlighted.

Corrected

The conclusion of the next - 3rd paragraph - of p 2 (spring fires) is not directly evident from climate change (annual data)

Corrected

In section 2, a solid and comprehensive description of the methodology is missing (also including a plot). It is good that the work builds on previous works done by the authors.

Corrected

In section 2 figures are overwhelming  text and the authors should move to supplement as many as they provide duplicated info. Two elements that should be described:

  1. uncertainty of the equations / factors
  2. are the emissions refer to annual estimates? (because this is mentioned many times in the paper)

1. Since the equations are widely used and linear and they are not based on statistical data, so we cannot reflect their range of accuracy.

2. Based on formula 4, we calculated total emission (TE) for a particular ecosystem type over the entire observation period, and the annual emission was determined as a long-term average

Is there any possibility of areas being burned more than once in a year and how is treated by the methodology of the authors?

Yes, this is possible, so we mapped spring and autumn fires separately.

When more than one E.O images are collected have the authors proceed with a comparison of the values?

As the authors indicated in the paper, all available images for the season were used to determinate the total fire area for the season.

Do annual results (e.g. fig 4) provide any evidence of climate change related impacts?

Only the areas of fires were considered by years, and no significant climate changes were revealed.

Section 3.2 tables are too many without providing extra info.

There is only one table in this section

Fig 5 contains many red colors - confusing.

Corrected

Fig 8 too complex Рис 8 слишком сложный.

Corrected

Again section4 carbon emission refer to annual values? this is not evident

To calculate the emission index the authors used the areas of fires in the ecosystems of Russia for 2020 (according to Greenpeace) and emission indicators per unit area

Table 5 belongs to supplement.

The table is provided to compare the results with other types of ecosystems from different regions. The authors decided to leave the table to the discretion of the editors of the journal.

Sincerely, authors

Reviewer 2 Report

Thank you for addressing the previous comments. The paper is much improved.

Three minor things.

Line 15: state the number of years.

Line 50: Source?

Line 186: Move to paragraph above.

Author Response

Dear reviewer!

Thank you very much for your comments. Please find below our responses to each of your comments.

Line 40: attention paid to good effect? Problem of forest fire solved?

Corrected

Line 53: Which two seasons?

Corrected

Line 77: agricultural development?

Corrected

Line 132: Lone paragraph. Method is the one mentioned in previous paragraph?

Corrected

Line 135-138. Hard to follow. Merge into above to form one paragraph.

Corrected

Line 150: palustrine is relatively uncommon term, maybe add that this refers to marsh land.

Corrected. We changed on mire.

Line 172: More description is needed of how exactly burned area was determined. Even if relying on well worn methods, they need a bit more description here.

Corrected

Line 196: What is “expert method”?

Corrected, it is “visual method”

Line 210: Extend table so that everything is in one main table and not spilt into two.

Then it takes up a lot of space, we would like to leave it at the discretion of the editors of the Journal

Line 220: 1,317.4% of the total land? Looks like an error.

No, this means that the territory burned out many times during the observation period

Line 223: Yes, looks like some interesting lag dynamics here.

-

Line 242: This looks like it belongs in methods.

Corrected

Line 299: This paragraph belongs in methods.

We think this paragraph belongs to the Discussion.

Line 376: Not really clear what the official statistics are from the text. Please clarify.

Corrected

Sincerely,

authors

Reviewer 3 Report

The purpose of this study is listed as “to estimate carbon emissions from fires in non-forest ecosystems of the MAL (within Khabarovsk Territory of the Russian Federation) based on Earth remote sensing data and materials from expeditionary studies.”  L95-97.  However, reading the abstract (L21-23) it appears the purpose of the study is to compare the emissions from non-forest fires to forest fires, to determine the importance of non-forest fires in the MAL area.  Please reconsider what is the purpose of this study.

Question: mires also occur in temperate areas also, not just boreal.  What is the definition of boreal that you are following to call this the boreal belt?  The land cover map [47] does not define boreal.  The reason this is important is because in the conclusion is the statement that emissions from humid meadows of the boreal belt are not well studied.  There are quite an increasing number of studies but they may be called temperate wetlands studies.  The information in this study is still useful especially in terms of the area burned, but it would be helpful to understand boreal vs temperate.

A commonly accepted approach is used to estimate estimates, which is to multiply area of fires by the carbon per area loss, and then sum over time.

The text indicates the novel part of the work is the improvement on estimating areas burned, that (L384-385) “The results of this study show that official statistics significantly underestimate the areas of meadow fires [46].” And then L402-404: “The influence of meadow fires is underestimated primarily due to inadequacies in the methods used to automatically map burned areas, leading to an underestimation of their areas.”  However, this new finding from this study is not highlighted in the abstract.  Please consider highlighting this finding in the abstract.  If the typical historic approach used is underestimating fires, then it is important to highlight the newer techniques should be used to reduce uncertainty.

L158-164:  This study is directed at non-forest ecosystems so it is unclear why the FC (carbon fraction) of 50% is used to convert biomass to carbon.  The citations [20,31] focus on forests, and 50% is the fraction used for wood.  But this study seems to be on some woody shrubs perhaps, and non woody plants.  A 2016 study in the MAL in DOI: 10.1134/S1067413615060089,  Kopoteva and Kuptsova, Russian Journal of Ecology, “Effect of Fires on the Functioning of Phytocenoses of Peat Bogs in the Middle-Amur Lowland” found used “To translate the plant matter (phytomass and production) into carbon (C), we used an overall average conversion coefficient of 0.45 (Titlyanova et al., 2005).”  In addition, the Intergovernmental Panel on Climate Change, in the National Greenhouse Gas Inventories Task Force, has quite a few emission factors (here in this study called specific carbon emissions) From https://doi.org/10.1016/j.heliyon.2019.e02329, it says “The carbon stock (carbon content) for the dry biomass of herbs and  litters is 47% of the total dry biomass of the quadrant (IPCC, 2007).”

L176:  I am not familiar with this use of the word “rags”.   Perhaps the word “swaths” is meant?

Table 4: please use names of ecosystem types instead of numbers, or include footnotes in the table that lists what the numbers mean.  Also, abbreviations used in Tables should be listed out in footnotes to the table.  Tables and Figures should stand alone.

L288-291: no it cannot be seen from Table 4 as written, which is why labeling the ecosystem types is so important.  Given this summary sentence, it is important to also label forest ecosystem types as forest so the reader can draw the same conclusion.

Table 5, please include what the Total emissions Tg are – are these Carbon emissions too?

Table 6.  Some of the footnote labels are letters not numbers (such as c in the row on forest).  Please update.

L382-384: actually there are a number of new studies on fires in the humid meadows of the boreal belt or perhaps temperate wetlands in various countries.

Minor grammar:

L144: change required to requires.

L258:  Figure 7 is listed but it seems to be referencing Figure 6.  Please check.

Author Response

Dear reviewer!

We would like to express our deep gratitude for your attention to our article. Your comments are allowed us to improve the paper.

“..The purpose of this study is to estimate carbon emissions from fires in non-forest ecosystems of the MAL (Khabarovsk Territory of the Russian Federation) based on Earth remote sensing data and materials from expeditionary studies.”  (p20, mid page)

This study would be more understandable and have more impact if it focused on the purpose, using one set of terminology throughout.  Please decide on terminology and stick with it.  For example, there is a discussion about the Khabarovsk Territory, which is actually the fourth largest state in the Russian Federation, and a map showing the Sanjiang-Middle Amur Plain (Figure 1) which includes an area in China, but this study focuses only on the Russian Lower Amur Lowland.  Is the Lowland the same as the Plain?  How much of the Middle Amur Plain does the Lower Amur Lowland occupy?   Why is the Khabarovsk Territory mentioned so much when the Amur Plain or Lowland only a small part of it?

You are right. We have made the corrections in the paper according your comments.

Another example, page 3 indicates “Currently, the total area of disturbed land is 7.6%, reflecting the low degree of development in the plain territory.”  With all the fires in this area, much of the land sounds disturbed, but only 7.6% is developed.  Does developed include agricultural lands? 

Improved.

Figure 3 uses anthropogenic lands (residential, agricultural), but it is unclear what developed means.  Is the “plain territory”  the same as the lowland area?   Why all these different terms?  Please stick with one definition and focus on that.  

You are right. We have made the corrections in the paper according your comments.

Another example: bottom of page 3. “To analyze the scale of pyrogenic impact on various ecosystems, the MAL ecosystem map developed by the authors [55] was used. Within the study area, four types of relief have been identified: floodplain, plain, foothill and low-mountain.”  However, Figure 2b is a map of ecosystems which differ from the map in Figure 3, and neither of these figures discuss ‘relief’.   Please provide definitions of the ecosystem types.  What is a meadow and what is a meadow-mire for example?   What would I look at to define the landscape as one or the other?  Why is Figure 2b shown if the authors are using definitions in Figure 3?  

You are right. We have made the corrections in the paper according your comments.

And Figure 1, which shows these areas broadly have different green colors but there is no legend to indicate what the dark green means in relation to the light green.  I assume the blue is water but there is no legend to confirm this.  In Figure 1, it looks like the whitish line is the border between China and the Russian Federation but this is not labeled.  

You are right. We have made the corrections in the paper according your comments.

While I am looking at Figures, Figure 2A has no legend for light and dark purple, yellow, green, and other colors either.

You are right. We have made the corrections in the paper according your comments.

Moving ahead with figures, in Figure 5  the color choice for the legend looks like the water of river part of the MAL has burned a number of times too.  The legend does not include a zero, which I assume would be white.

You are right. We have made the corrections in the paper according your comments.

Has the “anthropogenic lands” displayed in Figure 3 been overlain over the map shown in Figure 5, and labeled a “zero” (shown as white)? Or does Figure 5 strictly a compilation from Landsat 5, 7, and 8 as listed in the abstract?

As for Figure 5 we estimated the areas affected by fires without anthropogenic lands.

Another example of confusing terms, bottom page 5 to top of page 6 refers to MODIS (Terra and Aqua satellites) and VIIRS (NPP), but on page 6 this reads as though only Landsat was used.  Why is MODIS and VIIRS discussed if only Landsat is used (see Table 1)?

Here we consider existing approaches and justify the choice of the second approach based on the analysis of burnt areas according to Landsat data.

Then Figure 7 discusses  Visible Infrared Imaging Radiometer Suite (VIIRS) on the entire territory again (please focus on your study area of MAL if that is the study area and quit jumping between study areas or call the one study area the same name).

You are right. We have made the corrections in the paper according your comments.

And then a discussion of Figure 8 includes the Sentinel satellite (though it is mis-spelled as (Sentinnel-2).Also, Figure 8 shows just how little we can believe any of the fire areas, given how much they differ.   What is the difference between the A and B maps in Figure 8? I can somewhat see the difference with the magenta color but not understanding where any grey is for Greenpeace.

You are right. We have made the corrections in the paper according your comments.

Figure 5 shows spring and autumn fires, but the Figure 7 discussion mentions the summer fire season.  What months are included as spring; what as autumn; what as summer; etc?

You are right. We have made the corrections in the paper according your comments.

Page 8, under section 3.2, please describe the statistical design for the field studies and do not cite some other research proposal project.   The project is acknowledged in the back section; there is no reason to mention it here.  What is needed is a clear description of the sample design using the same definitions throughout.

You are right. We have made the corrections in the paper according your comments.

Page 14 near the top notes the anthropogenic nature of wildfires within the MAL, including agricultural land use, but the abstract on page 1 in the second sentence says  ”Nevertheless, ecosystems in which natural fires also make a significant contribution to anthropogenic CO2 emissions are poorly studied.”  Can you tell which fires are naturally caused fires versus those many fires that were caused by people?  What is a natural fire?   Is this study about natural fires versus fires caused by people (anthropogenic)?  Please clarify.

You are right. We have made the corrections in the paper according your comments. We consider “natural” or wildfires as fires on natural territories  (ecosystems).

In terms of the methodology used, it is not clear why on page 5,  that a “specific emission” (SE) was calculated by adding together meadows, forests and palustrine communities, when this study is focused on non-forest areas.  To focus on non-forest communities, do not add in forested areas!   Why there is now even more types of ecosystem areas, combining the 5 areas into 3, seems designed to confuse.   Also, using the abbreviation “SE” is confusing because normally SE means standard error in a research paper.  It is unclear why an “SE” needed to be calculated anyway instead of using one emission factor for each ecosystem type.

You are right. We have made the corrections in the paper according your comments. In addition, forest areas are considered to compare emissions from forest and non-forest fires, and due to the spatial heterogeneity (mosaicity) of vegetation cover they are components of complex ecosystem types (we took the spatial heterogeneity (mosaicity) of vegetation cover into account in the calculations).

Areas that repeatedly burn will likely have less surface fuels than those which burned once or twice over the long time period.  When samples were taken, did the design allow for samples relating to areas where it was estimated how often they burned?

We have made the corrections in the paper according your comments. The cycle of accumulation of combustible materials in ecosystems is 2-4 years. When planning the study, we selected points with various burns of different age.

Not including estimates of soil loss in non-forest is a major lacking in a study given all the research that has been published.   For example,  see  Fire frequency drives decadal changes in soil carbon and nitrogen and ecosystem productivity, Pellegrini et al, Nature, 2018, doi:10.1038/nature24668

Thank you very much for valuable reference. In this paper, we evaluated the direct emission of CO2 from wildfires. In addition, as we noted, considered fires within the study area take place in the spring, when the soils are still in a frozen state (no or minimal soil loss), or in autumn, when the territory is watered (no or minimal soil loss). We have made the corrections in the paper according your comments.

I do understand that with the area of land in each ecosystem type, one could multiply that area by an emission factor, per year, over time, and sum them.   This would be a standard approach to use.  What is hard to understand in this study is why the areas of ecosystems types are thrown together and not kept separate, and why soil carbon is not included in non-forest areas burned in particular.  Especially in a mire or wetland soil.

We calculated the average annual emission from different types of ecosystems due to the large interannual differences in the areas of fires including their distribution in different ecosystems,

The high mosaicity of the spatial structure of the territory's ecosystems makes impossible to clearly distinguish them into separate types. For example, within the MAL, 6% of the territory is occupied by herb, reed grass, sedge meadows in combination with herb mires (in depressions) and oak-white-birch-aspen belt forests (in elevated areas). They cannot be divided into separate categories according to remote sensing data. This required additional accounting of the areas of certain types of vegetation in the previously identified types of ecosystems based on aerial photography data from the Pfantom 4 UAV (for each description plot during fieldwork).

The very last sentence of this study is an informative sentence that points out why non-forest studies have not occurred as much as forest fire studies: “The influence of meadow fires is underestimated primarily due to inadequacies in the methods used to automatically map burned areas, leading to an underestimation of their areas.”   And the influence on fires on emissions are also underestimated in this type of study because soil carbon emissions were ignored.  This study uses carbon emissions, which I took to mean CO2 only type emissions, though emissions from fires will include non-CO2 emissions as well.

You are right. Indeed, the problem of estimating natural fires in various ecosystems has been known for a long time and has been studied by many authors. However, we believe that is precisely the lack of data on the true extent and frequency of fires in meadow and meadow-mire ecosystems of the boreal belt led to an underestimation of their contribution to pyrogenic carbon emission.

In Figure 6, the label is t/ga but I think t/ha  is meant.

Corrected.

We would hope that corrections made according your comment have led to substantial improvements of the manuscript.

Authors.

Round 2

Reviewer 3 Report

I wonder if an old version of the manuscript was posted, because the authors say they corrected Table 6, but it is not corrected.  There are still superscripts of b and c but only numbers are given an explanation.  Please correct Table 6 so it is clear what the superscript letters mean.

In terms of the lengthy explanation in the author’s response for the choice of using 0.5 for the conversion of biomass to carbon (line 166), please condense to a few sentences and citations and include the explanation in the manuscript.  As the manuscript is written, it appears the authors are choosing 0.5 from cited literature which is on forests, instead of 0.47 which is in the literature for non forest vegetation.  The explanation in a response to reviewer’s comments makes it clear the authors know better but are choosing 0.5 anyway. Please include several sentences and citations that were placed in comments to reviewer into the manuscript.  Thank you.

Figure 2, please explain in the legend what the small dots are indicating in areas in this figure.  Are the areas of darker red indicating multiple fires over the time frame?  Is the information provided by a small landscape or smaller watershed and that is why some areas look darker than others?  It is unclear how to interpret this figure.  Going to the website cited, it looks as if the authors may have downloaded layers from that site over the years specified, and simply plotted the summation of all the layers but this is not discussed in the manuscript.  Thank you.

Typo: line 301 needs to be fixed, some words are missing

Author Response

Dear reviewer!

Thank you very much for your comments. Please find below our responses to each of your comments.

I wonder if an old version of the manuscript was posted, because the authors say they corrected Table 6, but it is not corrected.  There are still superscripts of b and c but only numbers are given an explanation.  Please correct Table 6 so it is clear what the superscript letters mean.

Sorry for the confusion. Corrected

In terms of the lengthy explanation in the author’s response for the choice of using 0.5 for the conversion of biomass to carbon (line 166), please condense to a few sentences and citations and include the explanation in the manuscript.  As the manuscript is written, it appears the authors are choosing 0.5 from cited literature which is on forests, instead of 0.47 which is in the literature for non forest vegetation.  The explanation in a response to reviewer’s comments makes it clear the authors know better but are choosing 0.5 anyway. Please include several sentences and citations that were placed in comments to reviewer into the manuscript.  Thank you.

Thank you very much for your help. We added this information.

Figure 2, please explain in the legend what the small dots are indicating in areas in this figure.  Are the areas of darker red indicating multiple fires over the time frame?  Is the information provided by a small landscape or smaller watershed and that is why some areas look darker than others?  It is unclear how to interpret this figure.  Going to the website cited, it looks as if the authors may have downloaded layers from that site over the years specified, and simply plotted the summation of all the layers but this is not discussed in the manuscript.  Thank you.

We corrected the figure 2. There was a technical effects of  failures of software of the tool used

Typo: line 301 needs to be fixed, some words are missing

Corrected

With kind wishes,

Authors

This manuscript is a resubmission of an earlier submission. The following is a list of the peer review reports and author responses from that submission.

Round 1

Reviewer 1 Report

The presented manuscript by the authors deals with a timely analysis of carbon emissions within non-forest ecosystems. The topic it self has not been given adequate attention in recent years.

the introduction contains a wealth of references, but putting 53 references in a single paragraph does allow an accurate assessment of their relevance and critical findings.

The most important paragraph of Section 1, 2nd in p 2, is necessary to be revised so that innovative elements of the work are highlighted.

The conclusion of the next - 3rd paragraph - of p 2 (spring fires) is not directly evident from climate change (annual data)

in section 2, a solid and comprehensive description of the methodology is missing (also including a plot). It is good that the work builds on previous works done by the authors.

in section 2 figures are overwhelming  text and the authors should move to supplement as many as they provide duplicated info. Two elements that should be described:

  1. uncertainty of the equations / factors
  2. are the emissions refer to annual estimates? (because this is mentioned many times in the paper)

is there any possibility of areas being burned more than once in a year and how is treated by the methodology of the authors?

when more than one E.O images are collected have the authors proceed with a comparison of the values?

do annual results (e.g. fig 4) provide any evidence of climate change related impacts?

Section 3.2 tables are too many without providing extra info. Fig 5 contains many red colors - confusing. Fig 8 too complex

again section4 carbon emission refer to annual values? this is not evident

table 5 belongs to supplement.

Reviewer 2 Report

I tried inserting line #s myself but it didn't work.

The authors present an interesting study which helps improve understanding of how fire affects carbon dynamics of cover meadow and meadow-mire ecosystems in the Amur River ecoregion. The paper overall is well presented and organized. There are just a few places where methods need to made clearer.

Line 40: attention paid to good effect? Problem of forest fire solved?

Line 53: Which two seasons?

Line 77: agricultural development?

Line 132: Lone paragraph. Method is the one mentioned in previous paragraph?

Line 135-138. Hard to follow. Merge into above to form one paragraph.

Line 150: palustrine is relatively uncommon term, maybe add that this refers to marsh land.

Line 172: More description is needed of how exactly burned area was determined. Even if relying on well worn methods, they need a bit more description here.

Line 196: What is “expert method”?

Line 210: Extend table so that everything is in one main table and not spilt into two.

Line 220: 1,317.4% of the total land? Looks like an error.

Line 223: Yes, looks like some interesting lag dynamics here.

Line 242: This looks like it belongs in methods.

Line 299: This paragraph belongs in methods.

Line 376: Not really clear what the official statistics are from the text. Please clarify.

Reviewer 3 Report

“..The purpose of this study is to estimate carbon emissions from fires in non-forest ecosystems of the MAL (Khabarovsk Territory of the Russian Federation) based on Earth remote sensing data and materials from expeditionary studies.”  (p20, mid page)

This study would be more understandable and have more impact if it focused on the purpose, using one set of terminology throughout.  Please decide on terminology and stick with it.  For example, there is a discussion about the Khabarovsk Territory, which is actually the fourth largest state in the Russian Federation, and a map showing the Sanjiang-Middle Amur Plain (Figure 1) which includes an area in China, but this study focuses only on the Russian Lower Amur Lowland.  Is the Lowland the same as the Plain?  How much of the Middle Amur Plain does the Lower Amur Lowland occupy?   Why is the Khabarovsk Territory mentioned so much when the Amur Plain or Lowland only a small part of it?

Another example, page 3 indicates “Currently, the total area of disturbed land is 7.6%, reflecting the low degree of development in the plain territory.”  With all the fires in this area, much of the land sounds disturbed, but only 7.6% is developed.  Does developed include agricultural lands?  Figure 3 uses anthropogenic lands (residential, agricultural), but it is unclear what developed means.  Is the “plain territory”  the same as the lowland area?   Why all these different terms?  Please stick with one definition and focus on that.   Another example: bottom of page 3. “To analyze the scale of pyrogenic impact on various ecosystems, the MAL ecosystem map developed by the authors [55] was used. Within the study area, four types of relief have been identified: floodplain, plain, foothill and low-mountain.”  However, Figure 2b is a map of ecosystems which differ from the map in Figure 3, and neither of these figures discuss ‘relief’.   Please provide definitions of the ecosystem types.  What is a meadow and what is a meadow-mire for example?   What would I look at to define the landscape as one or the other?  Why is Figure 2b shown if the authors are using definitions in Figure 3?  

And Figure 1, which shows these areas broadly have different green colors but there is no legend to indicate what the dark green means in relation to the light green.  I assume the blue is water but there is no legend to confirm this.  In Figure 1, it looks like the whitish line is the border between China and the Russian Federation but this is not labeled.  While I am looking at Figures, Figure 2A has no legend for light and dark purple, yellow, green, and other colors either.  Moving ahead with figures, in Figure 5  the color choice for the legend looks like the water of river part of the MAL has burned a number of times too.  The legend does not include a zero, which I assume would be white.  Has the “anthropogenic lands” displayed in Figure 3 been overlain over the map shown in Figure 5, and labeled a “zero” (shown as white)?   Or does Figure 5 strictly a compilation from Landsat 5, 7, and 8 as listed in the abstract? 

Another example of confusing terms, bottom page 5 to top of page 6 refers to MODIS (Terra and Aqua satellites) and VIIRS (NPP), but on page 6 this reads as though only Landsat was used.  Why is MODIS and VIIRS discussed if only Landsat is used (see Table 1)?  Then Figure 7 discusses  Visible Infrared Imaging Radiometer Suite (VIIRS) on the entire territory again (please focus on your study area of MAL if that is the study area and quit jumping between study areas or call the one study area the same name).   And then a discussion of Figure 8 includes the Sentinel satellite (though it is mis-spelled as (Sentinnel-2).  Also, Figure 8 shows just how little we can believe any of the fire areas, given how much they differ.   What is the difference between the A and B maps in Figure 8? I can somewhat see the difference with the magenta color but not understanding where any grey is for Greenpeace.

Figure 5 shows spring and autumn fires, but the Figure 7 discussion mentions the summer fire season.  What months are included as spring; what as autumn; what as summer; etc?

Page 8, under section 3.2, please describe the statistical design for the field studies and do not cite some other research proposal project.   The project is acknowledged in the back section; there is no reason to mention it here.  What is needed is a clear description of the sample design using the same definitions throughout.

Page 14 near the top notes the anthropogenic nature of wildfires within the MAL, including agricultural land use, but the abstract on page 1 in the second sentence says  ”Nevertheless, ecosystems in which natural fires also make a significant contribution to anthropogenic CO2 emissions are poorly studied.”  Can you tell which fires are naturally caused fires versus those many fires that were caused by people?  What is a natural fire?   Is this study about natural fires versus fires caused by people (anthropogenic)?  Please clarify.

In terms of the methodology used, it is not clear why on page 5,  that a “specific emission” (SE) was calculated by adding together meadows, forests and palustrine communities, when this study is focused on non-forest areas.  To focus on non-forest communities, do not add in forested areas!   Why there is now even more types of ecosystem areas, combining the 5 areas into 3, seems designed to confuse.   Also, using the abbreviation “SE” is confusing because normally SE means standard error in a research paper.  It is unclear why an “SE” needed to be calculated anyway instead of using one emission factor for each ecosystem type.

Areas that repeatedly burn will likely have less surface fuels than those which burned once or twice over the long time period.  When samples were taken, did the design allow for samples relating to areas where it was estimated how often they burned?

Not including estimates of soil loss in non-forest is a major lacking in a study given all the research that has been published.   For example,  see  Fire frequency drives decadal changes in soil carbon and nitrogen and ecosystem productivity, Pellegrini et al, Nature, 2018, doi:10.1038/nature24668

I do understand that with the area of land in each ecosystem type, one could multiply that area by an emission factor, per year, over time, and sum them.   This would be a standard approach to use.  What is hard to understand in this study is why the areas of ecosystems types are thrown together and not kept separate, and why soil carbon is not included in non-forest areas burned in particular.  Especially in a mire or wetland soil.

The very last sentence of this study is an informative sentence that points out why non-forest studies have not occurred as much as forest fire studies: “The influence of meadow fires is underestimated primarily due to inadequacies in the methods used to automatically map burned areas, leading to an underestimation of their areas.”   And the influence on fires on emissions are also underestimated in this type of study because soil carbon emissions were ignored.  This study uses carbon emissions, which I took to mean CO2 only type emissions, though emissions from fires will include non-CO2 emissions as well.

This manuscript needs much clarification and editing to stay focused on one set of terminologies and study areas.

Typo:

In Figure 6, the label is t/ga but I think t/ha  is meant.